

# Experiencing fear during the pandemic: validation of the fear of COVID-19 scale in Polish

Irena Pilch[1], Zofia Kurasz[1] and Agnieszka Turska-Kawa[2]

[1] Institute of Psychology, University of Silesia, Katowice, Poland
[2] Institute of Political Science, University of Silesia, Katowice, Poland

Corresponding author
Irena Pilch, irena.pilch@us.edu.pl

## ABSTRACT

**Background:** The Fear of COVID-19 Scale (FCV-19S) is a brief self-report measure developed at the beginning of the coronavirus pandemic. The scale evaluates the emotional responses to COVID-19. To date, the FCV-19S has been translated and validated in about 20 languages and has been used in many published research. The current study aimed to validate the Polish version of the FCV-19S.

**Method:** The FCV-19S was translated into Polish using forward- and back-translation. From May 15th to June 15th 2020, a total of 708 community members (Sample 1: 383 persons, 209 females, Sample 2: 325 persons, 198 females) participated in the online study. The participants were recruited using online advertisements in social media. Participation was anonymous, voluntary, and without compensation. A confirmatory factor analysis was performed to test the one-factor structure of the FCV-19S. Then, measurement invariance was analyzed across samples, gender and age groups. To assess the validity of the scale, correlations between the FCV-19S and the remaining scales were computed.

**Results:** Internal consistency of the FCV-19S was good in both samples (Cronbach's alpha 0.89 and 0.85). The CFA showed that the one-factor model fits the data well (RMSEA = 0.067, 90% CI [0.059–0.094], CFI = 0.977, TLI = 0.965, GFI = 0.986). The criteria for configural, metric, scalar and strict invariance were met for all models tested. The FCV-19S scores correlated significantly with age, subjective vulnerability to the disease, neuroticism, self-reported compliance with the pandemic measures, and three kinds of preventive behavior (i.e., social distancing, hand hygiene, and disinfecting things).

**Conclusion:** The Polish version of the FCV-19S had a unidimensional structure, good reliability, and correlated as predicted with other variables. With the FCV-19S and the obtained results, healthcare professionals, researchers, and the government can gain more valuable information about people who may be at risk for negative psychological outcomes during the pandemic or who are not implementing protective behavior. The tool can be used in hospitals to quickly screen the level of fear in patients and minimize its severe adverse consequences.

## INTRODUCTION

At the end of 2019, the attention of most countries in the world turned to Wuhan in China, due to the increasing number of people infected by and dying from the novel coronavirus (Severe Acute Respiratory Syndrome Coronavirus 2; SARS-CoV-2). The virus has spread at a tremendous pace, causing a global health crisis. Infections have been confirmed in most countries in the world. The *World Health Organization (2020)* declared it to be a global pandemic several months later (in March 2020). The pandemic situation is a surprise and a challenge for everyone. In many countries, the pandemic takes a heavy toll and affects all aspects of people's lives, from private to work. Many occupational groups, such as healthcare professionals, were overwhelmed and exhausted from working for many hours, not to mention the fact that their work carried an enormous risk of contracting the virus. Additionally, limitation of work opportunities and a high rate of unemployment in other professional groups, financial consequences and the economic crisis had undoubtedly a negative impact on the psychophysical condition of citizens.

In Poland, the first case of COVID-19 was diagnosed on March 4[th]. One week later, social distancing measures were introduced. The whole country started operating in lockdown from March 14[th]. Poland closed its borders to non-citizens. On March 20[th], "epidemic state" was introduced. Schools, universities, offices, restaurants, and stores in shopping centers were closed. Social gatherings and non-essential movements were forbidden.

During the pandemic, most attention of medical care has been paid to infection control and methods of treating the physiological symptoms of the disease (*Dong & Bouey, 2020*; *Wang et al., 2020*). Mental health and well-being were relegated to the background, especially in the early stages of the pandemic (*Xiang et al., 2020*), also due to the lack of appropriate tools to assess the growing anxiety related to COVID-19. Currently, many people require professional psychological intervention. Prolonged focus on increased infection and mortality rates, grief over the loss of loved ones, and uncertainty about the future resulted in clinically significant psychological symptoms in many participants, i.e., increased levels of fear (*Ornell et al., 2020*; *Ahorsu et al., 2020b*; *Lin, 2020*), anxiety (*Peteet, 2020*), hopelessness (*Shaw, 2020*), irrational and unclear thoughts (*Ahorsu et al., 2020b*), or adjustment disorders (*Zhang et al., 2020*). *Tian et al. (2020)* in their study on the group of 1,060 Chinese people found that more than 70% of the participants had moderate and higher levels of psychological symptoms, including obsessive-compulsive disorder, interpersonal sensitivity, phobic anxiety, and psychoticism.

As noted by *Colizzi et al. (2020)*, the fear of COVID-19 is one of the core factors increasing the level of stress, exacerbating pre-existing mental health problems, and eliciting extreme anxiety reactions in time of the pandemic. This kind of fear can have many causes such as fear of getting infected, or of infecting loved ones, of being isolated from people and of contacting accidentally somebody who is infected (*Lin, 2020*). Thus, the proper evaluation of fear of COVID-19 is crucial. Emotions such as fear and anxiety can influence immunity, thus leading to and exacerbating disease state. What is also important is that emotional responses to threats can influence individuals' behaviors

during the pandemic. It has been proven that the level of fear has a significant impact on protective behavior during the pandemic. Additionally, elevated anxiety may lead to panic and misinterpretation of insignificant complaints as symptoms of the disease (*Asmundson & Taylor, 2020*). In turn, too little anxiety of viral infection is associated with a weaker tendency to follow the rules of hygiene and maintain social distance (*Taylor, 2019*). Thus, there is an urgent need for conducting research to understand the complexities of human behavior during the epidemic.

In that context, it is of crucial importance to have a reliable tool available in every language to examine the levels of fear related to COVID-19 in the society. There are a number of self-report scales that measure anxiety as a state or trait. However, these scales are not suitable to assess a person's response to specific stimuli. To measure emotional responses in the pandemic situation, some new instruments have been recently developed. One of them is the Fear of COVID-19 Scale (FCV-19S; *Ahorsu et al., 2020b*) which has gained a lot of attention as a tool with robust psychometric properties. The main aim of the current study was to validate the Polish version of this scale.

## Fear of COVID-19 scale

The Fear of COVID-19 Scale (FCV-19S) was originally developed for the general population in Iran to evaluate the emotional response to COVID-19 and it was published in the English version (*Ahorsu et al., 2020b*). Subsequently, this scale has been translated from English and validated in many languages, i.e., Arabic (*Alyami et al., 2020*), Bangla (*Sakib et al., 2020*), Brazilian Portuguese (*Faro et al., 2020*), English (*Perz, Lang & Harrington, 2020*; *Winter et al., 2020*), French (*Mailliez, Griffiths & Carre, 2020*), Italian (*Soraci et al., 2020*), Malay (*Pang et al., 2020*), Persian (*Ahorsu et al., 2020b*), Spanish (*Barrios et al., 2020*; *Huarcaya-Victoria et al., 2020*; *Martínez-Lorca et al., 2020*), Tamil (*Bharatharaj et al., 2020*), Turkish (*Haktanir, Seki & Dilmaç, 2020*; *Satici et al., 2020*), Urdu (*Mahmood, Jafree & Qureshi, 2020*), Chinese (*Chi et al., 2021*), Hebrew (*Bitan et al., 2020*), Japanese (*Masuyama, Shinkawa & Kubo, 2020*), Russian-Belarusian (*Reznik et al., 2020*), Romanian (*Stănculescu, 2021*), and Taiwanese (*Chang et al., 2020*). Previous scale validations covered many internally varied and different in size groups, e.g., 629 adolescents from two junior high schools in Japan (*Masuyama, Shinkawa & Kubo, 2020*), 1,700 participants aged 10–57 in China (*Chi et al., 2021*), or 693 members of the general Saudi population who were at least 18 years of age (*Alyami et al., 2020*).

The FCV-19S has been used in a number of published research on the relationships in terms of such important psychological issues as depression, anxiety and stress (*Ahorsu et al., 2020b*; *Perz, Lang & Harrington, 2020*; *Bitan et al., 2020*; *Alyami et al., 2020*), mental wellbeing (*Ahmed et al., 2020*; *Winter et al., 2020*), specific phobia(s) (*Soraci et al., 2020*), satisfaction with life (*Satici et al., 2020*), psychological distress (*Alyami et al., 2020*), post-traumatic stress disorder (*Huarcaya-Victoria et al., 2020*), generalized anxiety disorder (*Tsipropoulou et al., 2020*), resilience, self-compassion (*Chi et al., 2021*), perceived vulnerability to disease (*Masuyama, Shinkawa & Kubo, 2020*), preventive behavior during the pandemic (*Ahorsu et al., 2020a*), germ aversion, or even political beliefs (*Winter et al., 2020*).

Considering that COVID-19 is still in a development phase, it seems to be of crucial importance to identify individuals showing an excessive level of fear with the use of a reliable tool. For this purpose, the presented study aims at developing the Polish translation and validation of that instrument to assess the fear of COVID-19. The paper also analyzes the relationships between the fear of COVID-19 and preventive behaviors during the pandemic. With the FCV-19S and the obtained results, healthcare professionals, researchers, and the government can gain more valuable information about people who may be at risk for negative psychological outcomes from the pandemic or who are not implementing protective behavior. The tool can be used in hospitals to quickly screen the level of fear in patients and minimize its severe adverse consequences (*Pakpour, Griffiths & Lin, 2020*).

## Current study

The current study aimed to validate the Polish version of the FCV-19S. In the beginning, after the procedure of back-translation, we examined the structure of the scale and its psychometric properties. The original one-factor structure of the scale and measurement invariance across samples, gender and age groups were tested. Then, the relationships between the fear of COVID-19 and a set of its potential sociodemographic and psychological covariates (i.e., age and gender, perceived vulnerability to the coronavirus infection, and personality traits) were established to assess validity of the measure.

It was predicted that the fear of COVID-19 would be positively related to age (Hypothesis 1), because age is one of the important predictors of a severe course of the disease and in past studies it was positively related to the FCV-19S scores (*De Leo & Trabucchi, 2020*; *Meng et al., 2020*). We also predicted the positive relationship between subjective vulnerability to COVID-19 and the FCV-19S scores (Hypothesis 2), because the person's conviction of being or not being exposed to a severe course of the COVID-19 disease could induce more (or less) fear (*Yıldırım & Güler, 2021*; see also *Milne, Sheeran & Orbell, 2000*; *Carpenter, 2010*). As gender differences were obtained in other studies (the FCV-19S scores were higher in women than in men; *Broche-Pérez et al., 2020*), we expected to replicate this result (Hypothesis 3).

Among personality traits, neuroticism was found to be associated with the FCV-19S scores (*Kroencke et al., 2020*). Neuroticism (i.e., emotional lability and stress reactivity) is a trait which predisposes people to experience negative affect, and, more specifically, fear and anxiety. It has been postulated that the more neuroticism predominates as a central aspect of personality, the more threat will be perceived in a variety of situations (*Reynaud et al., 2012*). Negative emotional reactions in individuals high in neuroticism are typically triggered by stressors such as threat and uncertainty (*Lahey, 2009*; *Friedman, 2000*). Such emotional feelings are the bread and butter of pandemic times. Studies carried out during the time of COVID-19 found that people high in neuroticism worried more about disease-related information to which they reacted with more negative affect (*Kroencke et al., 2020*; *Abdelrahman, 2020*; *Aschwanden et al., 2020*). Thus, a positive relationship between levels of neuroticism and the FCV-19S was anticipated (Hypothesis 4).

Lastly, the criterion validity of the FCV-19S was checked. It was expected that the fear of COVID-19 would be associated with preventive behavior during the pandemic. Fear and
anxiety are unpleasant emotions which have a negative impact on well-being and are strongly associated with psychopathology (e.g., *Alyami et al., 2020*). Prolonged experiencing of strong negative emotions, especially in individuals with risk factors, can easily lead to many maladaptive results, which are broadly described in the literature, also in the context of the pandemic (e.g., *Li et al., 2020*). However, from the evolutionary point of view, fear is adaptive as it mobilizes individuals to cope with threats (*Nesse, 1990*). Thus, during the pandemic, fear can be a factor which can reduce the propensity for risky behavior and boost people's motivation to protect themselves. Indeed, several studies gave evidence of the positive role of the fear of COVID-19 in taking preventive behavior (e.g., *Wise et al., 2020*). For example, a study by *Harper et al. (2020)* showed that the fear of COVID-19 was the only predictor of behavior change related to preventive behavior during the pandemic. Thus, taking into account the results of the past studies (*Ahorsu et al., 2020b*; *Blagov, 2020*; *Bogg & Milad, 2020*; *Zettler et al., 2020*), we hypothesized that the FCV-19S scores would be positively related to preventive behavior during the COVID-19 pandemic (Hypothesis 5).

## MATERIALS & METHODS

### Participants (*N* = 708)

The data from the two samples were used. It was motivated by the need to obtain a large sample to perform the CFA on the FCV-19S scores. The data from both samples were collected during the same phase of the epidemic (May and June 2020) and the same procedure was used. *Sample 1* consisted of 383 participants (209 females—54.6%; age *M* = 31.4, *SD* = 11.6) from the general Polish population. Seventy-eight persons (20.4%) declared that they thought they might be at high risk for complications while developing COVID-19. The participants differed in level of education (secondary—162 persons, Bachelor's degree—63 persons, Master's degree—158 persons), marital status (single—153 participants, in relationship—230 participants) and employment (employed—242 participants, not employed—141 participants).

*Sample 2* consisted of 325 persons (198 females—60.9%; age *M* = 35.4, *SD* = 12.8) from the general Polish population. Seventy-nine participants (24.3%) declared that they thought they might be at high risk for complications while developing COVID-19. The participants differed in level of education (secondary—94 participants, Bachelor's degree—47 participants, Master's degree—184 participants), marital status (single—104 participants, in relationship—195 participants, other—26 participants) and employment (employed—208 participants, not employed—117 participants).

### Instruments

#### Samples 1 and 2

*The Fear of COVID-19* was measured with the Polish version of the Fear of COVID–19 Scale (FCV-19S; *Ahorsu et al., 2020b*), a seven-item measure with answers on a five-point scale (one-strongly disagree, five-strongly agree). The questionnaire was translated from English into Polish independently by two bilingual (Polish and English) researchers, whose native language was Polish. The blind back-translation into English was done by another

bilingual (Polish and English) scholar who was not involved in the initial translation. Then, the back translation was compared with the English version of the FCV-19S by a professional translator, and it was confirmed that the meaning of the translated Polish version of the FCV-19S was congruent with the meaning of the English version.

*Subjective vulnerability to the COVID-19 infection* was measured with one item prepared for this study ("Assess your susceptibility to coronavirus infection (related to age, health status, etc.)"). The participants answered using a 101-point scale (zero-very low susceptibility, 100-very high susceptibility).

*Subjective health situation* in the context of the coronavirus pandemic was measured with the question "Do you think you are at high risk for complications if you develop COVID-19"? with the answers yes/not.

Sample 1

*Engagement in preventive behavior* during the pandemic was measured with four items prepared for this study by the authors: "I am trying to reduce the chance of being contracted with the coronavirus", " I put a lot of effort into ensuring the safety of myself and my loved ones during the pandemic", "I try to respect the recommendations of medical authorities regarding behavior during the pandemic", "I accept most of the bans and orders introduced by the authorities to stop the pandemic". The participants responded using a five-point scale (one-strongly disagree, five-strongly agree). The scores were averaged across the items (in the current study: $\alpha = 0.86$).

Sample 2

*Personality traits.* Personality traits were assessed using the short version of the Polish adaptation of the International Personality Item Pool-Big Five-20 questionnaire (IPIP-BFM-20; *Topolewska et al., 2014*). This 20-item instrument assesses the Big Five personality traits (in the current study: extraversion, $\alpha = 0.87$, agreeableness, $\alpha = 0.70$, conscientiousness, $\alpha = 0.76$, neuroticism, $\alpha = 0.75$, and intellect, $\alpha = 0.66$) with answers on a five-point scale (one-very inaccurate, five-very accurate).

*Preventive behaviors during the pandemic* were measured with three questions prepared for this study by the authors ("Do you maintain a distance/isolate yourself from others?", "Do you try to wash your hands more often?", "Do you disinfect objects, e.g., door handles, a smartphone?"). These questions are linked to three important recommendations during the pandemic: social distancing/isolation, hand hygiene, and environmental cleaning/disinfection. The participants answered the questions using a 101-point scale (zero-definitely not, 100-definitely yes). The scores on these three dimensions were used as indicators of engagement in preventive behavior.

## Procedure

The participants were recruited using online advertisements in social media (Facebook, institutional websites and Internet forums). The content of the advertisements included a link to our online study with psychological questionnaires. The two separate surveys were completed online from May 15[th] to June 15[th] 2020 (Survey 1) and from May 15[th] to June 1[st] 2020 (Survey 2). During that period, there were about 12,000 confirmed cases of the disease in Poland (about 30,000 cumulative cases), and about 1,250 cumulative

death cases. The country was operating in lockdown. However, gradual relaxation in lockdown started from the end of May.

Participation in the study was anonymous, voluntary, and without compensation. The aims of each study were presented to participants at the beginning of both surveys. They were also ensured that the survey was anonymous and no personal data were collected. Written informed consent was obtained from all participants electronically before data collection. They were asked to confirm acceptance for participation in the survey by ticking the boxes provided. After that, participants provided sociodemographic information and completed the questionnaires. Study 1 and Study 2 included several other measures not related to the current study. The data for the present study are available as Supplemental Material. Both surveys were approved by the Ethics Committee of the University of Silesia in Katowice (KEUS.34/04.2020, KEUS.35/04.2020).

## Statistical analyses

Before the analyses, the data were examined to check for missing values and normality. No missing data were found. The structure of the Polish version of the FCV-19 scale was investigated using confirmatory factor analysis (CFA). The CFA and invariance analyses were performed in the conjoint sample, using the JASP (version 0.14) software. The mean- and variance-adjusted weighted least squares (WLSMV) procedure that does not assume the data are normally distributed was used (see *Brown, 2006*). The method uses diagonally weighted least squares (DWLS) to estimate the model parameters, the full weight matrix to compute robust standard errors, and a mean- and variance-adjusted test statistics. This estimation method was used because the FCV-19S utilizes a five-point Likert-type scale, thus the scores can be treated as ordinal ones. To evaluate the fit of the model, the selected fit indices (which are not related to the sample size; *West, Taylor & Wu, 2012*) were used. The value of the root mean square error of approximation (RMSEA) below 0.08 was treated as representing a sufficient fit and below 0.05 as representing a good fit (*MacCallum, Browne & Sugawara, 1996*). The comparative fit index (CFI) value and the Ticker-Lewis index (TLI) value above 0.95 were interpreted as showing a good fit (*Hu & Bentler, 1999*; *Schumacker & Lomax, 2010*). Following the recommendations (*MacCallum, Browne & Sugawara, 1996*, p.137, *Curran et al., 2003*), the 90% confidence interval (CI) for the RMSEA was reported, because it enables testing hypotheses of close and not-close fit.

To test for measurement invariance across samples, gender, and age groups, a series of multiple group CFA analyses (MGCFA) was performed, in the sequence with increasingly more restrictions on the parameters. Configural invariance means that the groups have the same CFA structure. When metric invariance was checked, the factor loadings were constrained to be equal across the groups. For scalar invariance, the factor loadings and means of the items were constrained to be equal. For strict (i.e., construct-level) invariance, residual variances and residual covariances were additionally constrained to be equal across the groups. Following the recommendation of *Cheung & Rensvold (2002)* that a single standard should be used to test model fit and measurement invariance, the differences in the fit indices (i.e., RMSEA and CFI) for constrained and unconstrained

**Table 1 Item translation and item-total correlation for the FCV-19S.**

| Items | Item-total correlation (corrected) | | Cronbach's Alpha (if item deleted) | | Mean (SD) total | | Mean (SD) males | | Mean (SD) females | |
|---|---|---|---|---|---|---|---|---|---|---|
| | Sample 1 (N = 383) | Sample 2 (N = 325) | Sample 1 (N = 383) | Sample 2 (N = 325) | Sample 1 (N = 383) | Sample 2 (N = 325) | Sample 1 (N = 174) | Sample 2 (N = 127) | Sample 1 (N = 209) | Sample 2 (N = 198) |
| 1 Bardzo boję się koronawirusa (I am most afraid of coronavirus-19) | 0.70 | 0.63 | 0.87 | 0.82 | 2.60 (1.04) | 2.72 (1.15) | 2.35 (0.98) | 2.46 (1.15) | 2.80 (1.04) | 2.89 (1.12) |
| 2 Czuję się nieswojo gdy myślę o koronawirusie (It makes me uncomfortable to think about coronavirus-19) | 0.72 | 0.67 | 0.87 | 0.82 | 2.42 (1.10) | 2.72 (1.27) | 2.14 (1.03) | 2.31 (1.22) | 2.66 (1.11) | 2.98 (1.24) |
| 3 Moje dłonie stają się wilgotne gdy myślę o koronawirusie (My hands become clammy when I think about coronavirus-19) | 0.66 | 0.53 | 0.88 | 0.84 | 1.52 (0.66) | 1.35 (0.72) | 1.36 (0.57) | 1.26 (0.62) | 1.67 (0.70) | 1.41 (0.77) |
| 4 Boję się, że stracę życie z powodu koronawirusa (I am afraid of losing my life because of coronavirus-19) | 0.72 | 0.64 | 0.87 | 0.82 | 1.95 (0.96) | 1.97 (1.11) | 1.84 (0.99) | 1.70 (0.92) | 2.05 (0.92) | 2.14 (1.19) |
| 5 Gdy oglądam wiadomości lub historie o koronawirusie w mediach społecznościowych, denerwuję się lub niepokoję (When watching news and stories about coronavirus-19 on social media, I become nervous or anxious) | 0.68 | 0.63 | 0.87 | 0.83 | 2.41 (1.12) | 2.46 (1.21) | 2.07 (1.97) | 2.13 (1.16) | 2.69 (1.08) | 2.67 (1.20) |
| 6 Nie mogę spać, gdyż martwię się że mogę złapać koronawirusa (I cannot sleep because I'm worrying about getting coronavirus-19) | 0.70 | 0.66 | 0.87 | 0.83 | 1.49 (0.72) | 1.36 (0.70) | 1.33 (0.65) | 1.23 (0.49) | 1.61 (0.75) | 1.45 (0.80) |
| 7 Serce wali mi jak młotem gdy myślę o zarażeniu się koronawirusem (My heart races or palpitates when I think about getting coronavirus-19) | 0.68 | 0.64 | 0.87 | 0.83 | 1.54 (0.77) | 1.35 (0.70) | 1.38 (0.73) | 1.20 (0.51) | 1.67 (0.77) | 1.45 (0.78) |
| The FCV-19 scale | – | – | 0.89 | 0.85 | 1.99 (0.71) | 1.99 (0.73) | 1.78 (0.65) | 1.76 (0.61) | 2.16 (0.72) | 2.14 (0.76) |

models were examined. Each model was compared to the baseline (or previous) model. The differences in the fit measures were analyzed. The criteria recommended by *Cheung & Rensvold (2002*, p. 251) and *Chen (2007*, p. 501) were applied: the hypothesis of invariance was supported when $\Delta$CFI $\leq -0.01$, $\Delta$ McDonald Fit Index $\leq -0.02$, and $\Delta$RMSEA $< 0.015$.

The remaining analyses, i.e., reliability analysis, correlations between the FCV-19S and other variables (the IPIP-BFM-20 subscales, susceptibility to COVID-19, age and engagement in preventive behavior), as well as testing for differences between independent groups (the Student t-test and the Mann–Whitney U test) were performed in both samples separately, with the use of the SPSS (version 26) software.

## RESULTS

### Internal consistency and descriptive statistics

The Polish translation of the items of the FCV-19S with their item-total correlations and Cronbach's alphas in both samples is shown in Table 1. Internal consistency of the FCV-19S was found to be good in both samples (alpha 0.89 and 0.85). The descriptive

statistics in Sample 1 and Sample 2 were as follow: M = 1.99, SD = 0.73, skewness 0.78, kurtosis 0.48 and M = 1.99, SD = 0.71, skewness 0.77, kurtosis 0.56, respectively.

## Dimensionality and measurement invariance

The CFA showed that a one-dimensional model has adequate fit ($\chi^2$ = 70.75, df = 14, $p < 0.001$ RMSEA = 0.067, 90% CI [0.059–0.094], CFI = 0.977, TLI = 0.965, GFI = 0.986). The chi-square test was significant, which is not desirable. However, this test is sensitive to sample size, meaning that significant results are usually obtained for larger samples (*Kline, 1998*). As our sample was large (N = 708), it could be the reason of this result.

Measurement invariance (configural, metric, scalar and strict) was tested across the study samples, gender (male, female) and age groups. The results are shown in Table 2. The MGCFA showed adequate fit for the one-factor model in both Sample 1 ($\chi^2$ = 26.18, df = 14, $p$ = 0.02, RMSEA = 0.05, CFI = 0.99, GFI = 0.99) and Sample 2 ($\chi^2$ = 15.72, df = 14, $p$ = 0.33, RMSEA = 0.02, CFI = 1.00, GFI = 0.99). The results for gender showed a good fit for the model in both male (N = 301, $\chi^2$ = 24.04 , df = 14, $p$ = 0.05, RMSEA = 0.05, CFI = 0.98, GFI = 0.98) and female (N = 407, $\chi^2$ = 53.6, df = 14, $p < 0.001$, RMSEA = 0.08, CFI = 0.97, GFI = 0.98) groups. The analyses performed in the two age groups separately also revealed the appropriate fit in younger (N = 358, $\chi^2$ = 37.64, df = 14, $p < 0.001$, RMSEA = 0.07, CFI = 0.98, GFI = 0.98) and older (N = 350, $\chi^2$ = 34.53, df = 14, $p < 0.001$, RMSEA = 0.06, CFI = 0.99, GFI = 0.99) groups. Thus, the criteria for configural invariance (i.e., one-factor structure) were met. The comparison of the relative fit of the nested models showed that also the criteria for metric invariance (i.e., invariant factor loadings), scalar invariance (i.e., invariant intercepts) and strict (construct-level) invariance were met for all models tested (see Table 2).

## Validity of the scale

As predicted, gender was associated with the FCV-19S scores in both samples. The fear of COVID-19 was significantly higher in women than in men (Sample 1, women M = 2.16 men M = 1.78, t = 5.39, df = 381, $p < 0.001$, Cohen's d = 0.55, 95% CI [0.35–0.76] ; Sample 2, women M = 2.14, men M = 1.76, t = 4.79, df = 323, $p < 0.001$, Cohen's d = 0.54, 95% CI [0.32–0.77]). There were also significant differences between sub-groups of people who declared belonging to the group of people at high risk for complications while developing COVID-19 (Sample 1, belonging: Mean rank = 236.0, not belonging Mean rank = 180.7, U = 8,462, $p < 0.001$, $\eta^2$ = 0.04; Sample 2, belonging: Mean rank = 200.8, not belonging Mean rank = 150.8, U = 6,729, $p < 0.001$, $\eta^2$ = 0.05). As predicted, the participants at high risk for complications had higher scores on the FCV-19S in both samples.

Zero-order correlations between the FCV-19S scores and age, subjective vulnerability to the disease, personality traits and protective behaviors during the pandemic are shown in Tables 3 and 4. As was hypothesized, the FCV-19S scores correlated positively with age (in Sample 1 r = 0.22, $p < 0.05$, in Sample 2 r = 0.19, $p < 0.05$) and subjective vulnerability (i.e., susceptibility) to COVID-19 in both samples (in Sample 1 r = 0.36, $p < 0.05$, in

**Table 2 Testing for measurement invariance across samples, gender and age groups.**

| Model | RMSEA | ΔRMSEA | CFI | ΔCFI | Invariance |
|---|---|---|---|---|---|
| *Sample invariance (Sample 1 vs. Sample 2)* | | | | | |
| Configural invariance | 0.061 | | 0.985 | | YES |
| Metric invariance | 0.059 | −0.002 | 0.983 | −0.002 | YES |
| Scalar invariance | 0.073 | 0.014 | 0.970 | −0.013 | YES/NO |
| Strict invariance | 0.077 | 0.004 | 0.960 | −0.010 | YES |
| *Gender invariance (males vs. females)* | | | | | |
| Configural invariance | 0.071 | | 0.978 | | YES |
| Metric invariance | 0.065 | −0.006 | 0.977 | −0.001 | YES |
| Scalar invariance | 0.060 | −0.005 | 0.977 | 0.000 | YES |
| Strict invariance | 0.058 | −0.002 | 0.974 | −0.003 | YES |
| *Age invariance (≤40 vs. >40)* | | | | | |
| Configural invariance | 0.072 | | 0.979 | | YES |
| Metric invariance | 0.077 | 0.005 | 0.970 | −0.009 | YES |
| Scalar invariance | 0.068 | −0.009 | 0.971 | 0.001 | YES |
| Strict invariance | 0.073 | 0.005 | 0.961 | −0.01 | YES |

**Table 3 Means, standard deviations and intercorrelations between study variables (Sample 2).**

| | M | SD | (1) | (2) | (3) |
|---|---|---|---|---|---|
| (1) Fear of COVID-19 | 1.99 | 0.71 | – | | |
| (2) Age | 31.37 | 11.60 | **0.220 (0.000)** | – | |
| (3) Vulnerability | 35.89 | 25.3 | **0.357 (0.000)** | **0.267 (0.000)** | – |
| (4) Engagement in preventive behavior | 3.94 | 0.85 | **0.335 (0.000)** | −0.028 (0.58) | **0.187 (0.000)** |

**Note:**
Significance level in parentheses. Significant correlation coefficients ($p < 0.05$) in bold. $N = 385$.

Sample 2 $r = 0.39$, $p < 0.05$). The correlations between personality traits and the fear of COVID-19 were also in line with the expectations. The FCV-19S scores correlated positively with Neuroticism ($r = 0.28$, $p < 0.05$) and there were no associations between the FCV-19S scores and Conscientiousness, Agreeableness or Extraversion. The fear of COVID-19 was also negatively related to Intellect ($r = −0.16$, $p < 0.05$), which was not anticipated. However, this relationship was weaker than the predicted one. Thus, the hypotheses H1-H4 received support.

The expected positive relationships between the FCV-19S scores and the declared preventive behaviors during the pandemic were obtained in both samples. In Sample 1, the FCV-19S scores correlated positively with the indicator of behavioral compliance with pandemic measures ($r = 0.34$, $p < 0.05$). In Sample 2, positive correlations between the FCV-19S and three kinds of behavior, i.e. social distancing ($r = 0.34$, $p < 0.05$), hand hygiene ($r = 0.29$, $p < 0.05$), and disinfecting things ($r = 0.36$, $p < 0.05$) were observed. These results supported Hypothesis 5.

**Table 4 Means, standard deviations and intercorrelations between study variables (Sample 1).**

| | M | SD | (1) | (2) | (3) | (4) | (5) | (6) | (7) | (8) | (9) | (10) |
|---|---|---|---|---|---|---|---|---|---|---|---|---|
| (1) Fear of COVID-19 | 1.99 | 0.73 | – | | | | | | | | | |
| (2) Age | 35.4 | 12.8 | **0.189** **(0.001)** | – | | | | | | | | |
| (3) Vulnerability | 30.5 | 24.9 | **0.389** **(0.000)** | **0.295** **(0.000)** | – | | | | | | | |
| (4) Agreeableness | 3.79 | 0.66 | 0.055 (0.331) | 0.071 (0.209) | 0.023 (0.681) | – | | | | | | |
| (5) Conscientiousness | 3.40 | 0.91 | 0.052 (0.363) | **0.196** **(0.001)** | 0.104 (0.068) | 0.070 (0.220) | – | | | | | |
| (6) Extraversion | 3.27 | 0.93 | −0.050 (0.379) | 0.037 (0.518) | −0.018 (0.757) | **0.250** **(0.000)** | **0.143** **(0.011)** | – | | | | |
| (7) Neuroticism | 2.82 | 0.84 | **0.282** **(0.000)** | **−0.182** **(0.001)** | 0.089 (0.116) | −0.015 (0.790) | −0.021 (0.706) | **−0.170** **(0.003)** | – | | | |
| (8) Intellect | 4.01 | 0.63 | **−0.157** **(0.006)** | −0.058 (0.306) | −0.060 (0.292) | **0.149** **(0.009)** | 0.098 (0.084) | **0.250** **(0.000)** | −0.110 (0.053) | – | | |
| (9) Social distancing/ isolation | 66.6 | 30.2 | **0.337** **(0.000)** | −0.024 (0.671) | **0.311** **(0.000)** | −0.038 (0.510) | −0.059 (0.301) | **−0.149** **(0.009)** | **0.138** **(0.015)** | 0.019 (0.737) | – | |
| (10) Hand hygiene | 82.7 | 25.3 | **0.288** **(0.000)** | 0.070 (0.209) | **0.198** **(0.000)** | 0.097 (0.089) | 0.026 (0.651) | 0.011 (0.853) | **0.137** **(0.016)** | −0.053 (0.354) | **0.422** **(0.000)** | – |
| (11) Disinfecting things | 55.1 | 0.36.7 | **0.360** **(0.000)** | 0.029 (0.605) | **0.196** **(0.000)** | **0.142** **(0.012)** | **0.127** **(0.025)** | −0.027 (0.635) | 0.064 (0.259) | 0.015 (0.791) | **0.365** **(0.000)** | **0.383** **(0.000)** |

**Note:**
Significance level in parentheses. Significant correlation coefficients ($p < 0.05$) in bold. $N = 325$.

## DISCUSSION

In this study, the structure and psychometric properties of the Polish version of the FCV-19S were examined. Moreover, relationships between the fear of COVID-19 and a set of socio-demographic and individual variables were established. The issue how to measure such emotions as fear during the pandemic seems very important as the COVID-19 pandemic is far from over. The FCV-19S is one of the questionnaires developed at the beginning of the pandemic crisis to measure negative emotions which can occur in this new and threatening situation. The measure is short and simple to use and has been translated into about 20 languages so far. The Polish version of this questionnaire showed a clear factorial structure and good psychometric properties. The one-factor model fits the data well.

The results of the analyses showed measurement invariance across samples, gender and age groups. The configural invariance means that the participants (from Sample 1 and Sample 2, men and women, older and younger) conceptualize the construct measured by the FCV-19S in the same way. Metric invariance signifies that the factor loadings are equal across these groups. Thus, cross-group comparisons can be performed. Scalar invariance means that the vectors of item intercepts are also invariant across the groups, which is important when the latent means are compared. Scalar invariance across the samples was only partially confirmed (ΔRMSEA was under the cut-off level), but it received full support for gender and age groups. Thus, it can be assumed that the criteria

for strong factorial invariance are met, which enables the comparison of latent means across gender and age groups (*Cheung & Rensvold, 2002*, p. 238). The hypothesis of construct-level (strict) invariance was also supported so that comparisons of correlations between the fear of COVID-19 and other variables are possible across the groups (see *Cheung & Rensvold, 2002*).

The results of the current study revealed positive relationships between the levels of the fear of COVID-19 and age, gender and subjective vulnerability to the disease. Among personality traits, the FCV-19S scores correlated with neuroticism. These relationships are congruent with our predictions. Therefore, the results of the current study provide evidence about the validity of the Polish version of the FCV-19S. It seems important for scientists and practitioners to predict which individuals can be prone to experience higher levels of fear of the coronavirus in the pandemic situation, as this kind of fear can have strong consequences, which are important from both individual and social points of view. Our results have shown that older individuals, women, people who feel vulnerable, and those higher in neuroticism can develop higher levels of the fear of COVID-19, with all the consequences it implies. These findings are in line with other studies (e.g., *De Carvalho, Pianowski & Gonçalves, 2020*).

Taking preventive actions which have the potential to slow the spread of an infectious disease seems particularly important during an epidemic. In the case of the COVID-19 pandemic, the lack of effective therapies, relatively high infectiousness, the presence of asymptomatic cases and the anticipated long duration of the pandemic make it even more important to make people adhere strictly to many pandemic rules. Negative emotions triggered by the pandemic situation can impact obedience. Correlation analysis revealed positive relationships between the FCV-19S scores and preventive behavior in both samples. The associations of the FCV-19S scores with the scores on the scale measuring engagement in preventive behavior during the pandemic (i.e., accepting preventive measures, respecting recommendations and trying to reduce the probability of infection) were found in Sample 1. The relationships between the fear of COVID-19 and taking preventive actions during the pandemic (i.e., social distancing/isolation, hand hygiene and cleaning/disinfecting) were obtained in Sample 2. Generally, people who felt more fearful about the coronavirus declared stronger engagement in preventive behavior. This result provides the initial evidence of the predictive value of the Polish version of the FCV-19S.

Our study has some limitations. First, our participants were relatively young. Therefore, the majority of them scored rather low on the FCV-19S, which could impact the results. Second, only self-report measures were used. Self-reports may not be the optimal source of data, particularly in the case of preventive behavior. Third, the cross-sectional study design makes drawing causal inferences difficult. Fourth, by using an online survey, our sample was restricted to people with access to the Internet. Nevertheless, the current data enable the conclusion that the Polish version of the fear of COVID-19 scale is a reliable tool for the measurement of fear of being contracted during the coronavirus pandemic.

## CONCLUSIONS

The main aim of the study was to examine the structure and psychometric properties of the Polish version of the FCV-19 scale. We tested the original structure and measurement invariance across samples, gender and age groups. The scale showed a clear one-factor structure and good psychometric properties. The fear of COVID-19 can have strong clinical consequences, so it is particularly important for scientists and practitioners to predict which individuals can be prone to experience it. Our results have shown that older individuals, women, people who feel vulnerable, and those higher in neuroticism can develop higher levels of the fear of COVID-19. Considering that participants in the study were relatively young, and the majority of them scored rather low on the FCV-19S, the experiences of fear of COVID-19 in the elderly require further investigation. Nevertheless, the obtained data enable the conclusion that the Polish version of the fear of COVID-19 scale is a reliable tool for the measurement of fear of being contracted during the coronavirus pandemic.

### Funding

Publication financed by the funds granted under the Research Excellence Initiative of the University of Silesia in Katowice. The funders had no role in study design, data collection and analysis, decision to publish, or preparation of the manuscript.

### Grant Disclosures

The following grant information was disclosed by the authors:
Research Excellence Initiative of the University of Silesia in Katowice.

### Competing Interests

The authors declare that they have no competing interests.

### Author Contributions

- Irena Pilch conceived and designed the experiments, performed the experiments, analyzed the data, prepared figures and/or tables, authored or reviewed drafts of the paper, and approved the final draft.
- Zofia Kurasz analyzed the data, authored or reviewed drafts of the paper, and approved the final draft.
- Agnieszka Turska-Kawa performed the experiments, authored or reviewed drafts of the paper, and approved the final draft.

### Human Ethics

The following information was supplied relating to ethical approvals (i.e., approving body and any reference numbers):

The Ethics Committee of the University of Silesia in Katowice granted ethical approval to carry out the research (KEUS.34/04.2020, KEUS.35/04.2020).

## Data Availability

The raw measurements and materials are available in the Supplemental File.

## Supplemental Information

Supplemental information for this article can be found online at http://dx.doi.org/10.7717/peerj.11263#supplemental-information.

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
