# Peer review of "Experiencing fear during the pandemic: validation of the fear of COVID-19 scale in Polish"

_PeerJ, doi:10.7717/peerj.11263_

## Round 0.1 · original submission · Minor Revisions

Please, address all the comments of the reviewers.

Reviewer 1 ·

Basic reporting

In general, the authors have done a good work. However, they may have to read through the manuscript once more for few typos and grammatical errors.
Also, authors may stick to the use of a research term. For instance, they may use either "participants" or "subjects" but not both. Further, they may revise the use of "sex" and "gender" in their manuscript.

Lines 274-278: Authors may try and make Table 1 more comprehensive by including the demographic characteristics and Cronbach's alpha.

At line 281, authors stated "...RMSEA=0.067, 90% CI [0.059, 0.094]..." Can authors explain why they used 90%CI as the others were all 95%?

Lines 309 - 322: Authors may state the r values related to the relationships.

Experimental design

Line 66: Participants:- this paragraph needs revision. Specifically,
Authors may need to justify the use of two different group of samples in this study.

Start new sentences with words (Seventy-eight persons) and not figures (78 persons) as in lines 168 and 174.

The sentences in lines 168-169 and 174-175 are not clear. Specifically, "...declared adherence to the group of people at high..." is not comprehensible.

Procedure
Authors stated "Some other data not related to this study and yet unpublished were included in the surveys...". Authors may have to explain why this data was added to this study especially as they were not collected together with the data of this study.

Validity of the findings

The conclusion is too lengthy. Authors may further summarize it into a paragraph.

Additional comments

No comment

Reviewer 2 ·

Basic reporting

The article is clear, references, context and lierature analysis is exaurient.

Experimental design

The research has conducted with rigurous investigations and Statistical Analysis. Technical & Ethical Standards were rispected.

Validity of the findings

The results are intersting and abolutly adapt to polish population. O

Additional comments

Very intersting work

Reviewer 3 ·

Basic reporting

it is acceptable

Experimental design

it is acceptable

Validity of the findings

it is acceptable

Additional comments

I have read the topic with interest.
I think that the study has some strengths: Considerable sample size, and working on hot topic. However, the manuscript needs some clarifications/modifications:
3- Abstract: please report sampling procedure and study time. I would see a clear message for implications for public.
4- Introduction: I think that it will be great if the authors can provide some information regarding how the Poland react to the COVID-19 (e.g., government' policies, infected cases, deaths). I believe that this will strengthen the manuscript. Please report study aims and
hypothesis. please update your references on the fear validations:
For example:
Chi, X., Chen, S., Chen, Y., Chen, D., Yu, Q., Guo, T., ... & Zou, L. (2020). Psychometric evaluation of the fear of COVID-19 scale among Chinese population.

Stănculescu, E. Fear of COVID-19 in Romania: Validation of the Romanian Version of the Fear of COVID-19 Scale Using Graded Response Model Analysis. International Journal of Mental Health and Addiction, 1-16.

Pakpour, A. H., Griffiths, M. D., & Lin, C. Y. (2020). Assessing psychological response to the COVID-19: the fear of COVID-19 Scale and the COVID Stress Scales. International Journal of Mental Health and Addiction.

Ahorsu, D. K., Imani, V., Lin, C. Y., Timpka, T., Broström, A., Updegraff, J. A., ... & Pakpour, A. H. (2020). Associations Between Fear of COVID-19, Mental Health, and Preventive Behaviours Across Pregnant Women and Husbands: An Actor-Partner Interdependence Modelling. International Journal of Mental Health and Addiction, 1-15.

Chang, K. C., Hou, W. L., Pakpour, A. H., Lin, C. Y., & Griffiths, M. D. (2020). Psychometric testing of three COVID-19-related scales among people with mental illness. International Journal of Mental Health and Addiction, 1-13.

5- Method: sampling procedure is still unclear to me. I would see more information on COVID-19 condition during data collection period.

---

## Round 0.2 · accepted · Accept

Please, I suggest to use the title "Experiencing fear during the pandemic: Validation of the Fear of COVID-19 scale in Polish”, because we prefer to have the study design in the title. If possible, please, some minor suggestions indicated by reviewer 4.

Thank you very much and congratulations!

Reviewer 4 ·

Basic reporting

NOTHING

Experimental design

NOTHING

Validity of the findings

Comments to the Author
It is an interesting research that contributes to understanding the role of fear in Poland population during the time of quarantine in COVID-19 pandemic. There is needed some valid and reliable instruments in order to assess the fear in this kind of studies about the self-perception of emotions, anxiety, fear and mental health problems.

The paper has been written well. It is very important that authors had obteined an ethical approval of the University of Silesia in Katowice. The paper is appropriate, the sample is adequate and it has current references. The authors confirm the good psychometric properties of the questionnaire as well as the original version and in other validations, traslate, countries and samples, for instance, university students.

I have minor revision. I have reported my comments in the following:
- Please add some information on mortality and morbidity rates on COVID-19 during data collection time.
- Why are there two samples? Both from the general Polish population? I don´t understand the objective of these.
- Which is the median score of fear in FCV-19S?
- Add more authors and studies in discussion about the psychometric properties in the different adaptation in all languages because now the paper has not much references and how do you explain this results?. Discussion has only 3 references, it is so poor. Please, add more and more. It is the most important step to find validity of the findings or not but authors should add more evidence about your results

·

Basic reporting

Professional article structure, figures, tables. Raw data shared.

Experimental design

Rigorous investigation performed to a high technical & ethical standard.

Validity of the findings

Local validation of an existing tool

Additional comments

This psychometric study evaluated the characteristics of the questionnaire to assess the fear of the epidemic. The questionnaire is already translated and used in many countries, so this study adds only one national version to the many that exist. The procedure adopted by the authors is the standard one. the results are in line with expectations. The authors reported and discussed the limitations of the study.